# Characterization of heart rate variability in end-stage renal disease patients after kidney transplantation with recurrence quantification analysis

**Amara Hazel Solorio-Rivera[1], Martín Calderón-Juárez [2,3,4] \*, Jesús Arellano-Martínez [1], Claudia Lerma [5], Gertrudis Hortensia González-Gómez [6] \***

1 Department of Nephrology, Hospital General "Dr. Miguel Silva", Morelia, Mexico, 2 Plan de Estudios Combinados en Medicina, Faculty of Medicine, Universidad Nacional Autónoma de México, Mexico City, Mexico, 3 International Collaboration on Repair Discoveries, Faculty of Medicine, University of British Columbia, Vancouver, BC, Canada, 4 Division of Physical Medicine and Rehabilitation, Faculty of Medicine, University of British Columbia, Vancouver, BC, Canada, 5 Department of Electromechanical Instrumentation, Instituto Nacional de Cardiología Ignacio Chávez, Mexico City, Mexico, 6 Department of Physics, Faculty of Sciences, Universidad Nacional Autónoma de México, Mexico City, Mexico

\* hortecgg@ciencias.unam.mx (GHGG); martin.cal.j@comunidad.unam.mx (MCJ)

**Data Availability Statement:** All relevant data are within the paper and its Supporting Information files.

## Abstract

Heart rate variability (HRV) is a noninvasive approach to studying the autonomic modulation of heart rate in experimental settings, such as active standing sympathetic stimulation. It is known that patients with end-stage renal disease during active standing have few changes in HRV dynamics, which are improved after hemodialysis. However, it is unknown whether the response to active standing is recovered after definitive treatment with kidney transplantation. This work aims to assess the change in HRV dynamics in the supine position and active standing through time and frequency-based metrics, as well as recurrence plot quantitative analysis (RQA). We studied HRV dynamics by obtaining 5-minute electrocardiographic recordings from kidney transplant recipients who underwent an active standing test. The mean duration of heartbeats and their standard deviation diminished in active standing, compared with the supine position. Also, the low-frequency component of HRV and the presence of diagonal and vertical structures in RQA were predominant. A larger estimated glomerular filtration rate was significantly correlated with broader HRV in the supine position and during active standing. The narrower HRV during active standing may indicate a sympathetic response to external stimuli, which is expected in a functional cardiovascular system, and may be influenced by renal function.

## Introduction

Heart rate variability (HRV) is the instantaneous change in heart rate. These heart rate variations are tightly related to respiratory and autonomic modulations of the cardiovascular system [1]. In end-stage renal disease (ESRD), patients who are treated with hemodialysis show

**Funding:** The author(s) received no specific funding for this work.

**Competing interests:** The authors have declared that no competing interests exist.

chronic sympathetic hyperactivity that produces decreased HRV modulation with a predominance of low-frequency oscillations and a blunted HRV response to active standing [2]. However, cardiac autonomic modulation and baroreflex sensitivity improve after renal transplantation [3].

Recurrence plots are a robust nonlinear analysis tool that can be implemented in HRV time series to observe their dynamic behavior [4]. This tool graphically represents the recurrences of a system to a particular dynamical state. Recurrence quantification analysis (RQA) has been previously used for the analysis of HRV in ESRD [5] during an active standing test (the change from a supine position to active standing). This maneuver allows studying the effects of blood volume redistribution and sympathetic stimulation of the cardiovascular system [6].

We have reported that ESRD patients (before treatment with hemodialysis) lack adjustment in HRV dynamics during active standing stimulation [2, 7] that may change after hemodialysis treatment. Although, HRV may improve after a kidney transplant [3], the response to sympathetic stimulation by active standing in transplant recipients remains unknown. The objective of this work is to assess the response of HRV linear indices and RQA measures in the active standing test on kidney transplant recipients.

## Materials and methods

### Study protocol and participants

We included 16 patients with kidney transplantation who were recruited during follow-up consultation in a second-level hospital in Mexico with a median age of 32 [27–35] years old and with equal number of males and females. All patients had a stable functional allograft at the time of recruitment with a median creatinine of 1.1 [0.8–1.4] mg/dL and an estimated glomerular filtration rate of 73.5 [58.2–89.5] mL/min/1.73m$^2$ (calculated with the equation proposed by Chronic Kidney Disease Epidemiology Collaboration 2021). Patients were treated with a triple immunosuppression regimen, using a calcineurin inhibitor, prednisone, and mycophenolate. Three patients were on antihypertensive treatment: one with amlodipine, another with losartan, and the last with metoprolol. Other clinical characteristics are shown in **Table 1**. During recruitment, patients were asked to avoid caffeinated beverages and other stimulants 24 hours before the evaluation. Electrocardiographic (ECG) recordings were obtained in a supine position during spontaneous breathing for 10 minutes. The recording continued for an additional 10 minutes after the patients stood up by themselves (active standing). The first half of the ECG recording in each position was discarded and the remaining 5 minutes were analyzed.

Our study followed the ethical standards of the 1964 Helsinki Declaration and the later amendments. The study protocol was approved by the Ethics Committee of the Hospital General "Dr. Miguel Silva" (protocol number 590/02/21). All patients provided their written informed consent.

### ECG preprocessing

One-channel ECG recordings were obtained with BioHarness 3.0 (Zephyr technology) at a sampling frequency of 250 samples per second. We performed R wave identification and visual supervision using Kubios HRV premium software. The identification of arrhythmias and artifacts in ECG recordings was visually supervised by a trained physician. Sporadic ventricular extrasystoles were found in two recordings, a correction algorithm was used to substitute extrasystoles [8]. Less than 5% of heartbeats were replaced, as recommended in the HRV analysis guidelines [1]. We named the final HRV time series as the NN intervals [1] (distance

**Table 1. Clinical characteristics of patients with end-stage renal disease after kidney transplant.** Continuous data are shown as median (interquartile range), and categorical variables are shown as absolute frequency (relative frequency).

| | |
|---|---|
| BMI (kg/m$^2$) | 26.3 (22.2–29.3) |
| Age (years) | 32 (27–35) |
| Male sex | 8 (50%) |
| Time since CKD diagnosis (months) | 108 (72–180) |
| Time since kidney transplant (months) | 62 (36–113) |
| Cellular graft rejection history | 5 (31.3%) |
| *CKD etiology* | |
| Diabetes mellitus | 2 (12.5%) |
| Hypertension | 5 (31.3%) |
| Unknown | 11 (68.8%) |
| *Type of kidney donator* | |
| Deceased donor | 7(43.8%) |
| Living donor | 9(56.3%) |
| *Immunosuppressive treatment* | |
| Tacrolimus | 11 (68.8%) |
| Cyclosporine | 5 (31.3%) |
| Leucocytes (x1,000/μL) | 6.4 (5.5–7.7) |
| Hemoglobin (g/dL) | 13.9 (12.6–15.4) |
| Hematocrit (%) | 42 (39.8–43.7) |
| Platelets (x1,000/μL) | 238.5 (218.5–291.0) |
| Glucose (mg/dL) | 90 (84–102) |
| Creatinine (mg/dL) | 1.1 (0.8–1.4) |
| BUN (mg/dL) | 22.3 (16.2–25.4) |
| Uric acid (mg/dL) | 6.3 (5.3–7.8) |
| eGFR (mL/min/1.73m$^2$) | 73.5 (58.2–89.5) |
| Sodium (mmol/L) | 141 (140–142) |
| Potassium (mmol/L) | 4.3 (4–4.5) |
| Chloride (mmol/L) | 108 (106–109) |
| Phosphorus (mmol/L) | 3 (2.5–3.4) |
| Calcium (mg/dL) | 9.4 (9.1–9.6) |

BMI: body mass index. BUN: blood urea nitrogen. eGFR: estimated glomerular filtration.

between one R wave and the next R wave in ECG recording), referring to the time between "normal" heartbeats.

## HRV measures

We calculated the mean of the NN time series (meanNN [s]), its standard deviation (SDNN [s]), the standard deviation of differences between adjacent NN intervals (SDSD [s]), and the percentage of adjacent NN intervals differing by more than 20 ms (pNN20 [%]) [1, 9]. To calculate frequency-based indices, we resampled the HRV time series at 3 Hz, then applied a 300 data points Hamming window with 50% overlapping. Finally, we used the Fast Fourier Transform to calculate the low-frequency band (LF– 0.04 to 0.15 Hz) and the high-frequency band (HF– 0.15 to 0.4 Hz). LF and HF are reported as normalized units (n.u.). Also, we calculated the LF/HF ratio [1].

### Recurrence quantitative analysis

We calculated RQA metrics using the "Cross Recurrence Plot" toolbox for MATLAB [10]. Embedding delay was chosen at the average mutual information function's first local minimum for each time series. The embedding dimension was calculated at the false nearest neighbors function's first local minimum. For the recurrence plot construction, we used the fixed amount of neighbors' norm (we set recurrence density to 0.07) [11]. We present diagonal line-based metrics determinism and Shannon's entropy, as well as the vertical line-based metrics of laminarity, and trapping time [4]. Also, the recurrence time type 1 (T1) and recurrence time type 2 (T2) were calculated.

### Statistical analysis

We report categorical variables as the absolute frequency (relative frequency), and continuous variables as the median (interquartile range). We compared HRV metrics in supine position vs active standing with Wilcoxon signed-rank test. Also, a nonparametric correlation between eGFR and HRV measures was evaluated with Pearson's correlation coefficients ($\rho$) with both, supine position and active standing, to increase the dynamic range in HRV. The statistical significance was set to $p < 0.05$.

## Results

**Fig 1** shows examples of time series, power spectral density (PSD), and recurrence plots of one subject in the supine position and active standing. The values of time domain indices, meanNN, SDNN, SDSD, and pNN20 are larger in the supine position than active standing (**Fig 2**).

**Fig 3** shows that LF is smaller in the supine position, and the HF value is larger in the supine position than during active standing. Accordingly, the LF/HF ratio is also smaller in the supine position.

Regarding RQA measures, determinism, laminarity, and trapping time are statistically smaller in the supine position (**Fig 4**). **Fig 5** shows that T1 is larger in the supine position, and T2 during active standing. We observed significant correlations between meanNN and other HRV measures (both positions, N = 32), linear (SDNN [$\rho = 0.538$, $p = 0.002$], SDSD [$\rho = 0.541$, $p = 0.001$], pNN20 [$\rho = 0.522$, $p = 0.002$], LF [$\rho = -0.396$, $p = 0.025$], HF [$\rho = 0.389$, $p = 0.028$], and LF/HF [$\rho = -0.393$, $p = 0.026$]), and nonlinear (Determinism [$\rho = -0.422$, $p = 0.016$], laminarity [$\rho = -0.506$, $p = 0.003$], trapping time [$\rho = -0.371$, $p = 0.037$], and T2 [$\rho = -0.481$, $p = 0.005$]). There was not a significant correlation between meanNN and T1 ($\rho = 0.163$, $p = 0.374$).

**Table 2** shows the correlation between eGFR and HRV measures in the supine and active standing. Few correlations are observed between eGFR and HRV indices (SDSD, pNN20, determinism, and Shannon's entropy) in the supine position. During active standing, significant correlations are found between eGFR and most of the HRV measures (SDNN, SDSD, pNN20, LF, HF, LF/HF, determinism, laminarity, trapping time, and T2).

## Discussion

This work shows the evaluation of HRV during active standing tests using traditional HRV indices and nonlinear RQA measures in kidney transplant recipients. We found a significant change in almost all HRV indices, as well as a predominance of vertical structures in HRV recurrence plots during active standing.

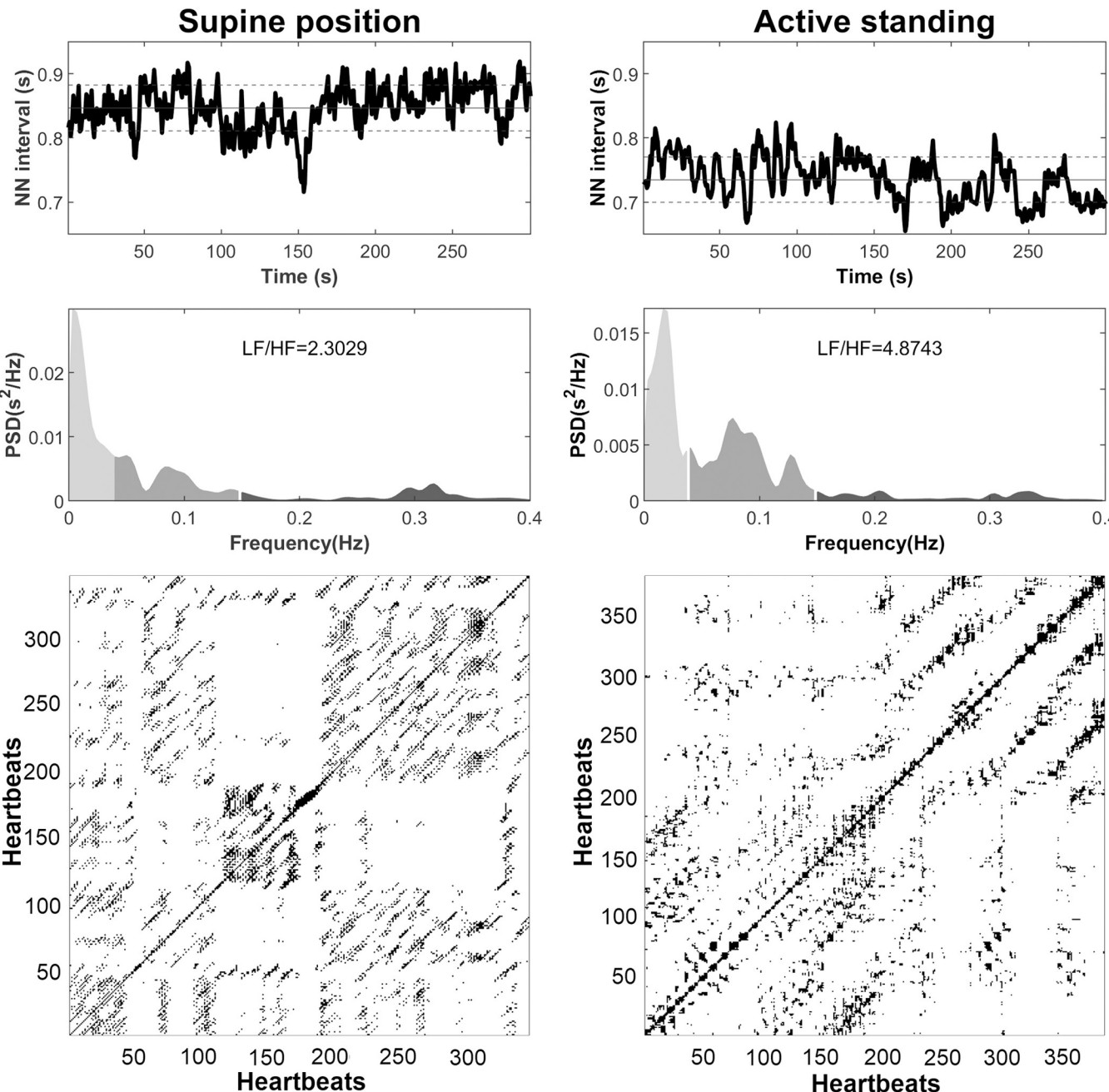

**Fig 1.** Heart rate variability (HRV) time series (upper panels) of one patient in the supine position and active standing, their respective power spectral density (PSD) (middle panels), and recurrence plots (bottom panels). Horizontal lines in the upper panels show the mean (solid horizontal line) and standard deviation (dashed horizontal line).

I other studies, it has been observed that the SDNN increases largely after kidney transplantation [12]. Furthermore, kidney transplant recipients without diabetes have been observed to have a larger standard deviation of 5-minute heartbeat intervals within a 24-hour period compared to recipients with diabetes [13]. To compare appropriately in this work, the potential improvement after kidney transplantation should be compared with the HRV indices in the same subjects before and after transplantation. Therefore, we focus on the comparisons between the supine position and active standing.

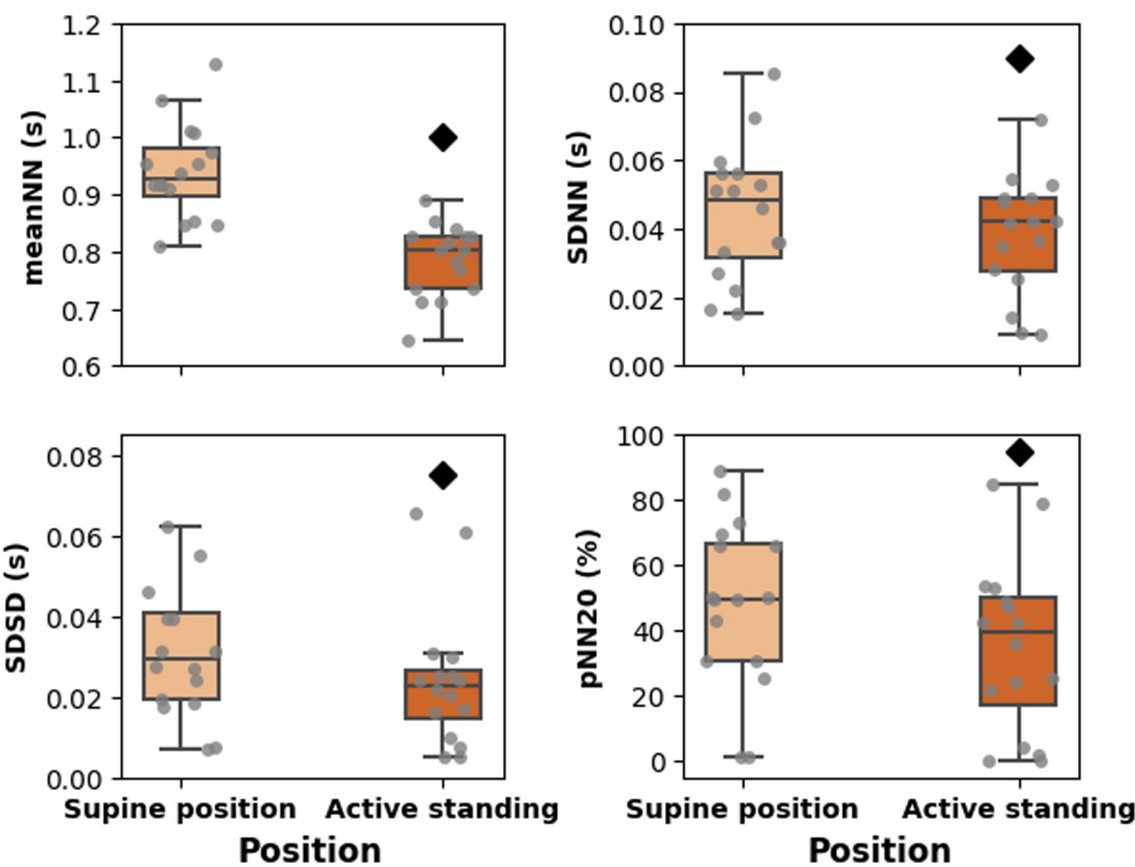

**Fig 2. Boxplots showing the values of linear (statistical) heart rate variability (HRV) indices during the active standing test.**

However, we briefly discuss the response to active standing stimulation in ESRD patients on hemodialysis treatment from the same clinic. In previous works, the SDNN of ESRD patients before hemodialysis treatment did not change significantly from a supine position to active standing [7]. In this study, patients after kidney transplantation display a significant decrease in SDNN after active standing, as expected in healthy subjects [13]. Regarding RQA measures, ESRD patients treated with hemodialysis do not show a statistically significant change in response to the active standing test (laminarity, trapping time, and T2) [7]. However, we found here that these measures increase in their value during active standing in transplant recipients.

It is expected that linear [1] and RQA measures [11, 13] change in healthy subjects in response to active standing, given that the rapid changes in heart rate dynamics are mainly attributed to an autonomic response [11]. The lack of a significant response in ESRD maybe is due to autonomic nervous system impairment (dysautonomia), as is supported by experimental and observational studies [3], although more factors could be involved. Due to the significant changes in HRV in response to active standing, it is possible that kidney transplantation improves the autonomic response [14]. Other authors have shown that renal transplantation improves cardiac sympathetic innervation by means of measuring 24-hour HRV and myocardial scintigraphy [15]. Furthermore, in previous work we observed significant correlations between meanNN and linear measures, as reported in healthy human subjects and other experimental scenarios [13, 16], whereas no correlation is found in patients with ESRD before hemodialysis, but such correlation is recovered after hemodialysis treatment.

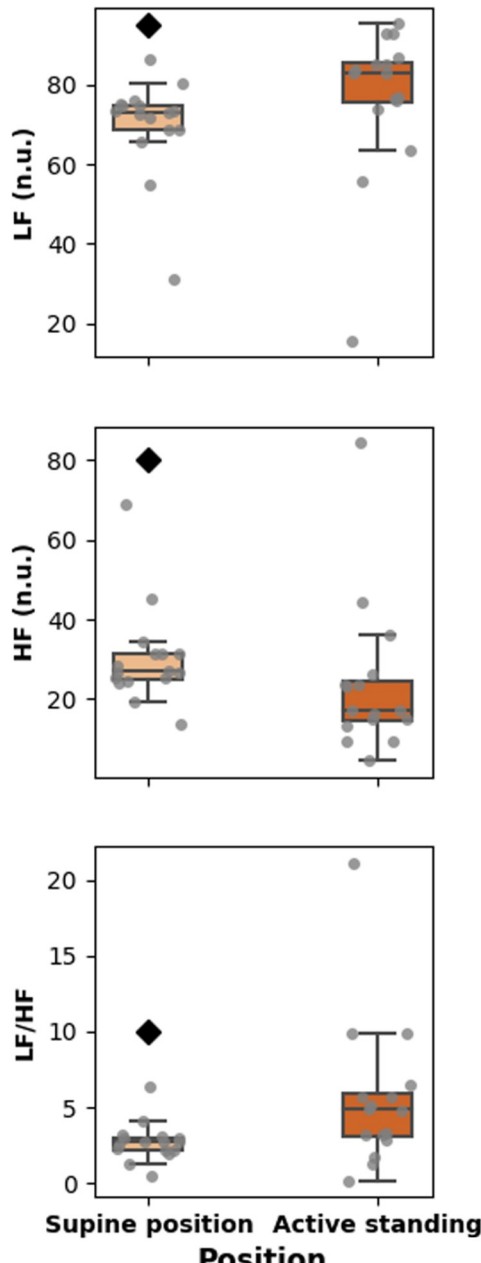

**Fig 3. Boxplots showing the values of linear (frequency-based) heart rate variability (HRV) indices during the active standing test.**

After renal transplantation, the cardiac output, arterial stiffness and baroreflex sensitivity improve [17], although mineral metabolism, renin-angiotensin-aldosterone disturbances [17] and inflammatory response of the host may continue to exist in these patients [18, 19]. All of the above factors may influence HRV [20–22] of this set of patients, but more extensive studies with a methodological approach for comparing data from the same patients before and after the kidney transplant should be conducted in the future to address their influence on the HRV. The patients included in our study are young (median age 32 years), which is

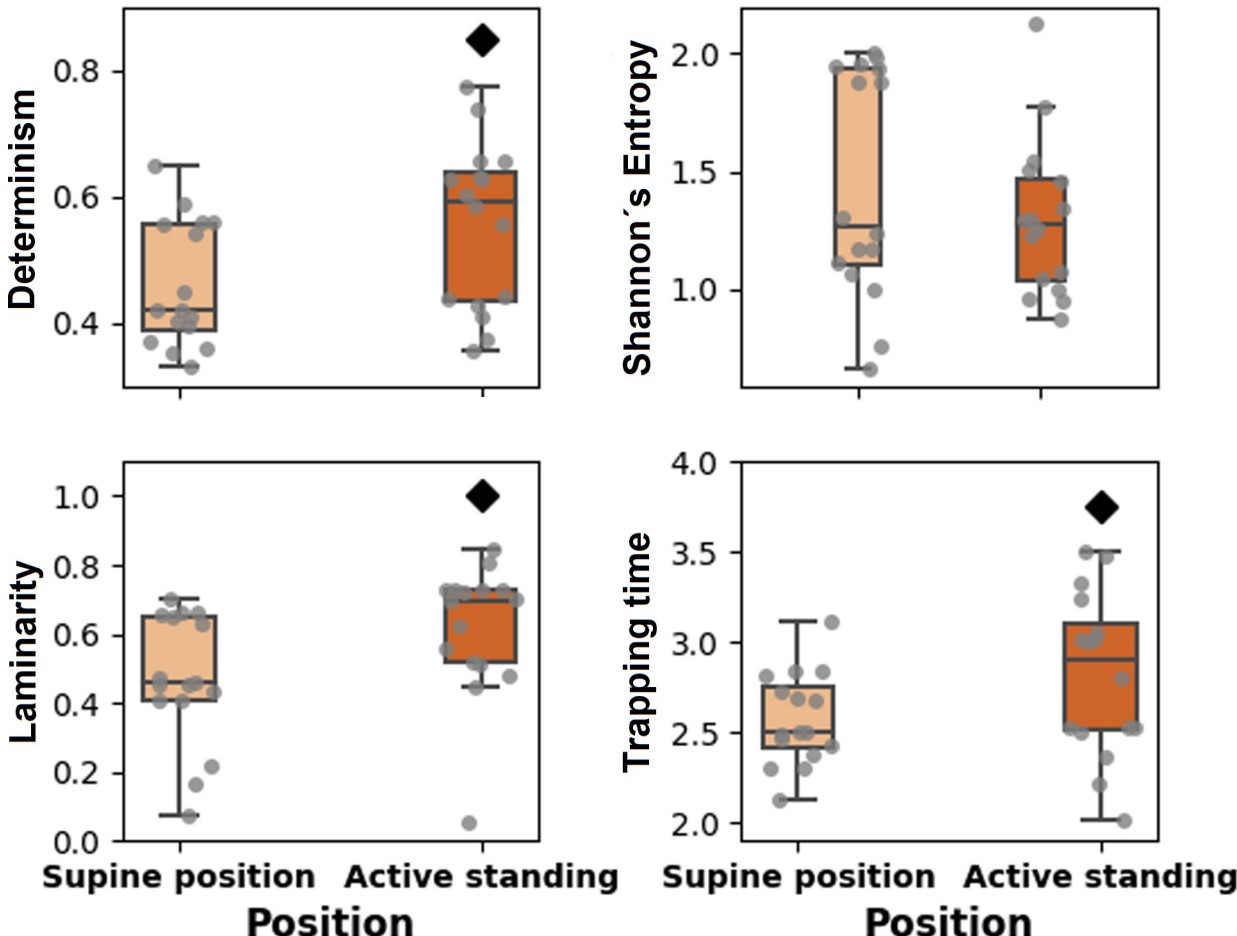

**Fig 4. Boxplots showing the values of recurrence quantification analysis (RQA) measures of heart rate variability (HRV) during the active standing test.**

representative of the age of most transplant recipients in our country, with a median age of 29 years [23]. We acknowledge that advanced age, combined with diabetes mellitus, which is the situation of most of our hemodialysis patients worldwide, is a risk factor for dysautonomia and therefore results in reduced variability in heart rate response to physiological challenges. A study specifically designed with this particular population would be necessary to understand the impact of kidney transplantation on HRV.

Moreover, our study does not have HRV measurements of these patients before kidney transplantation; hence, it cannot be determined whether kidney transplantation improves RQA measures. Nevertheless, based on a mathematical model designed to study the HRV and cardiovascular adaptations in end-stage renal disease [24], it is likely that the changes in RQA measures observed in active standing (e.g., determinism, Shannon's entropy, laminarity, and trapping time) may be the result of fluid overload, increased blood pressure and vascular resistance [5, 24]. These cardiovascular parameters are expected to improve (partially) with the restitution of kidney functions [17]. In this work, a greater global variability (SDNN) is related to higher eGFR, even during sympathetic stimulation by active standing. Also, a broader variability between consecutive heartbeats (SDSD, pNN20), lower LF predominance, and diminished laminar states (laminarity and trapping time) are linked to better renal allograft function

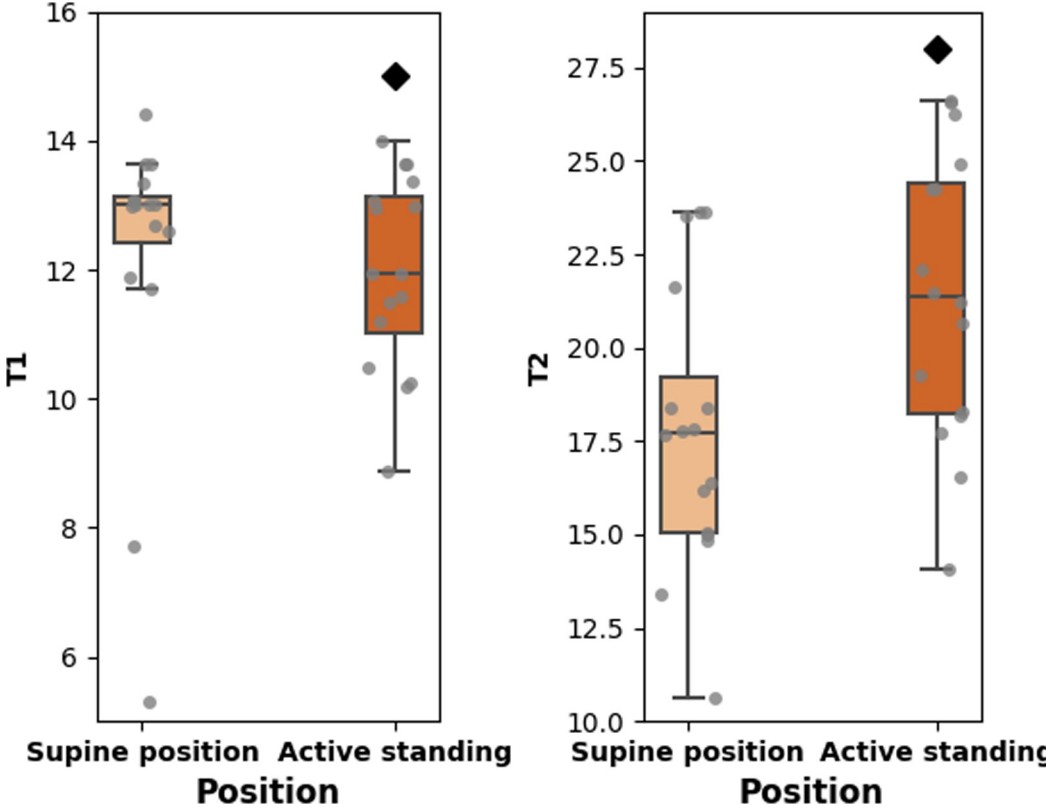

**Fig 5. Boxplots showing the values of T1 and T2 values of heart rate variability (HRV) during the active standing test.**

(eGFR); as well as larger trajectories in HRV dynamics (T2). Correlation between greater HRV and better eGFR has been observed in patients with chronic kidney disease in a broad population with different autonomic maneuvers [25]. A better eGFR may be linked to an improvement in HRV measures as observed in our work; however, larger studies that include assessment before kidney transplantation are needed to explore this relationship and to

**Table 2. Pearson's correlation coefficients (ρ) between estimated glomerular filtration rate (eGFR) and heart rate variability (HRV) indices in the supine position and active standing.**

|  | Supine position (ρ) | p | Active standing (ρ) | p |
|---|---|---|---|---|
| meanNN (s) | 0.406 | 0.118 | 0.414 | 0.111 |
| SDNN (s) | 0.427 | 0.099 | 0.517 | 0.04 |
| SDSD (s) | 0.566 | 0.022 | 0.743 | 0.001 |
| pNN20 (%) | 0.536 | 0.032 | 0.753 | 0.001 |
| LF (n.u.) | -0.430 | 0.096 | -0.517 | 0.040 |
| HF (n.u.) | 0.398 | 0.127 | 0.517 | 0.040 |
| LF/HF | -0.418 | 0.107 | -0.517 | 0.040 |
| Determinism | -0.568 | 0.022 | -0.581 | 0.018 |
| Shannon's Entropy | -0.566 | 0.022 | -0.472 | 0.065 |
| Laminarity | 0.141 | 0.601 | -0.643 | 0.007 |
| Trapping time | -0.501 | 0.048 | -0.655 | 0.006 |
| T1 | 0.409 | 0.115 | 0.272 | 0.308 |
| T2 | -0.180 | 0.506 | -0.714 | 0.002 |

determine whether this reflects an improved cardiovascular function. The eGFR is a complex renal function that is influenced by several clinical characteristics (e.g., sex, age, obesity, diabetes, hypertension), meat consumption and smoking [26]. Although we found a statistically significant correlation between HRV indices and eGFR, this might be influenced by other physiological factors, such as volume and electrolyte status [27], endocrine control (e.g., thyroid and parathyroid hormones) [28, 29] and antihypertensive medications [30]. In this exploratory study, such factors were not evaluated, and further research is needed to assess their specific role with HRV modulation and eGFR. Since our approach of non-linear HRV indices obtained by recurrence plot analysis during a physiological stimulus (active standing) has been useful in addressing the dynamic adaptability in ESRD patients treated by HD [5, 7], the current findings of correlation between eGFR and RQA measures could motivate future investigation about the adaptability of the cardiovascular control mechanisms in transplanted patients.

Among the myriad of HRV measures, such as entropy-based metrics, detrended fluctuation analysis, Poincaré plots, and symbolic dynamics [31, 32]; we employed RQA for broadening the previously known behavior of HRV from the recurrence plots perspective of ESRD to kidney transplantation. However, a wider characterization of nonlinear HRV metrics should be considered in future studies, and comparison the implemented maneuver in this work (active standing) with the gradual tilt test [31]. Furthermore, quantifying the presence of nonlinear information *per se* through surrogate data testing is often complicated by the nonstationarity in HRV [14]; this aspect falls outside the scope of this study and should be studied in larger populations.

## Conclusions

HRV of kidney recipients shows a diminished variability during active standing, compared to the supine position. The dynamic behavior of HRV also changed in response to active standing with a predominance of vertical structures in recurrence plots (measured through laminarity and trapping time). These changes suggest a marked sympathetic response to active standing, as well as a preserved capacity to respond to such orthostatic challenges with appropriate adjustments in the dynamic behavior of the cardiovascular control system. The wider HRV statistical dispersion, as well as the improved frequency-based and RQA measures, may be influenced by a better glomerular filtration rate. Studies with different methodological designs are required to demonstrate causality in the improvement of HRV in patients with ESRD through kidney transplantation.

## Supporting information

**S1 File. List of abbreviations.**
(DOCX)

**S1 Data.**
(CSV)

## Author Contributions

**Conceptualization:** Amara Hazel Solorio-Rivera, Martín Calderón-Juárez, Jesús Arellano-Martínez, Claudia Lerma, Gertrudis Hortensia González-Gómez.

**Data curation:** Amara Hazel Solorio-Rivera, Martín Calderón-Juárez.

**Formal analysis:** Martín Calderón-Juárez.

**Investigation:** Amara Hazel Solorio-Rivera, Martín Calderón-Juárez, Jesús Arellano-Martínez, Claudia Lerma, Gertrudis Hortensia González-Gómez.

**Methodology:** Martín Calderón-Juárez, Gertrudis Hortensia González-Gómez.

**Resources:** Claudia Lerma, Gertrudis Hortensia González-Gómez.

**Software:** Martín Calderón-Juárez.

**Supervision:** Martín Calderón-Juárez, Jesús Arellano-Martínez, Gertrudis Hortensia González-Gómez.

**Visualization:** Martín Calderón-Juárez, Gertrudis Hortensia González-Gómez.

**Writing – original draft:** Amara Hazel Solorio-Rivera, Martín Calderón-Juárez, Jesús Arellano-Martínez, Claudia Lerma, Gertrudis Hortensia González-Gómez.

**Writing – review & editing:** Amara Hazel Solorio-Rivera, Martín Calderón-Juárez, Jesús Arellano-Martínez, Claudia Lerma, Gertrudis Hortensia González-Gómez.

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
