## [Decision Letter · Decision Letter 0]

20 Mar 2023

PONE-D-23-02358Characterization of nonlinear heart rate variability of end-stage renal disease patients after kidney transplantationPLOS ONE

Dear Dr. González-Gómez,

Thank you for submitting your manuscript to PLOS ONE. After careful consideration, we feel that it has merit but does not fully meet PLOS ONE’s publication criteria as it currently stands. Therefore, we invite you to submit a revised version of the manuscript that addresses the points raised during the review process.

We look forward to receiving your revised manuscript.

Kind regards,

Keiko Hosohata, Ph.D.

Academic Editor

PLOS ONE

Reviewers' comments:

Reviewer's Responses to Questions

**Comments to the Author**

1. Is the manuscript technically sound, and do the data support the conclusions?

Reviewer #1: Yes

Reviewer #2: Partly

Reviewer #3: Yes

Reviewer #4: Yes

2. Has the statistical analysis been performed appropriately and rigorously? 

Reviewer #1: Yes

Reviewer #2: I Don't Know

Reviewer #3: Yes

Reviewer #4: Yes

3. Have the authors made all data underlying the findings in their manuscript fully available?

Reviewer #1: No

Reviewer #2: Yes

Reviewer #3: Yes

Reviewer #4: Yes

4. Is the manuscript presented in an intelligible fashion and written in standard English?

Reviewer #1: Yes

Reviewer #2: No

Reviewer #3: Yes

Reviewer #4: Yes

5. Review Comments to the Author

Reviewer #1: The study exploits recurrence quantification analysis (RQA) to characterize heart rate variability (HRV) of end-stage renal disease (ESRD) patients after kidney transplantation (KT).

The study is interesting and originally discuss HRV during orthostatic stimulus in ESRD patients after KT. However, some issues need a more profound discussion.

1) The major finding of the present study is the greater regularity of HRV during standing in ESRD patients after KT and this result supports the general improvement after KT. A greater regularity of HRV is compatible with an increased sympathetic modulation during orthostatic challenge as proven in A. Porta et al, J Appl Physiol, 103, 1143-1149, 2007. The link between loss of HRV complexity and postural challenge should be more deeply discussed and corroborated with adequate references.

2) It remains unclear why the authors used RQA instead of simpler nonlinear, and more widely utilized, descriptors of HRV irregularity such as markers based on conditional entropy.

3) The authors evoked the presence of nonlinear dynamics in HRV (e.g., in the title) but no tests were utilized to check the actual presence of nonlinear dynamics. Tests should be carried out to corroborate the choice of a nonlinear marker (see e.g., A. Porta et al, Front Physiol, 6, 71, 2015 for a procedure to test the presence of nonlinear dynamics). In alternative, linear markers of complexity are present (see A. Porta et al, IEEE Trans Biomed Eng, 64, 1287-1296, 2017).

4) More details should be given to better understand how the parameters of RQA were chosen. For example, it is unclear how the embedding dimension and the tolerance to assess pattern similarity were selected. Please clarify in the Methods.

5) The white squares in the recurrence plots in Fig.1 denote the presence of nonstationarities. Their presence is visible as changes of the mean in the same figure. Since stationarity is a prerequisite of RQA, it should be checked (see V. Magagnin et al, Physiol Meas, 32, 1775-1786, 2011 for a stationarity test). In alternative, the authors should discuss the impact of nonstationarities over the final conclusion of the study.

6) English language should be profoundly revised (e.g., the second phrase from the bottom of the Abstract). Construction of the phrases should be checked (e.g., the first phrase of the Introduction) as well as the usage of punctuation.

Reviewer #2: This paper concerns “Characterization of nonlinear heart rate variability of end-stage renal disease patients after kidney transplantation”

It is a struggle for me to read the paper because of many language issues and a huge number of abbreviations. I am not familiar with all these cardiology terms and abbreviations, so that does not help. Although the subject is interesting, after trying to read it several times, at present I cannot comment properly because of the numerous textual issues.

Table 2 abbreviations after normal text would make it more readable.

A list of abbreviations is also required

A clear legend of the figures is necessary

The following sentences need improvement.

1. However, it is unknown whether this response is recovered after definitive treatment with kidney transplantation.

2. We studied HRV dynamics by obtaining short-term ECG recordings from kidney transplant recipients that underwent active standing stimulation

3. Larger estimated glomerular filtration rate (eGFR) in significantly correlated with

broader gross HRV in supine position and during active standing. The loss in HRV during active standing may indicate a preserved sympathetic response to external stimul

4. is known that chronic sympathetic hyperactivity produces decreased HRV modulation with predominance of lowfrequency oscillations, and a blunted HRV response to active standing

5. This tool allows to graphically represent the recurrence of a system to a particular dynamical state

6. This maneuver allows studying of the effects of volume redistribution and sympathetic

stimulation of the cardiovascular system

7. However a kidney transplant is the treatment of choice for ESRD

8. We included 16 patients with ESRD after kidney transplantation that were recruited during followup consultation in a second-level hospital in Mexico with a median age of 32 �27-35� years old

9. and triple immunosuppression treatment with a calcineurin inhibitor plus prednisone and mycophenolate.

10. Subjects were asked to avoid caffeinated beverages and other stimulants

11. We name the final HRV time series as NN

12. We calculate the mean of NN time series (meanNN

13. Embedding dimension was calculated at the first local minimum reaching zero in the fixed amount of neighbors function

14. Categorical variables are reported as absolute frequency (relative frequency) and continuous

variables as median (interquartile range).

15. Time domain indices, meanNN, SDNN, SDSD, and pNN20 are larger during supine position in comparison to active standing

16. Few correlations are observed in supine position (SDSD, pNN20, DET and ENT), but during active standing a significant correlation is found between eGFR and most of HRVmeasures (SDNN, SDSD, Pnn20, LF, HF, LF/HF, DET, LAM, TT and T2).

17. Also, a larger standard deviation of 5-minutes intervals in 24 hours (SDANN) of kidney transplant recipients without diabetes than those recipients without diabetes

18. Here, patients after kidney transplantation

19. However, we found here that in transplant recipients, these measures increase their value during active standing.

20. change in healthy subjects in response to active standing. given that the rapid changes in heart rate

21. is due to dysautonomia. This

22. other experimental scenarios [14, 18], which are not found in patients with ESRD but after HD treatment.

23. approach such be conducted

24. our study do not have

25. it is likely that the changes in RQA measures in active standing (e.g. DET, ENT, LAM and TT) may be a result of fluid overload

26. resistance [5, 25], these

27. As we show in this

28. is related to better eGFR

29. to a improve in HRV

30. to explore this relation and determine

31. in comparison with supine position

32. a consistent predominance of vertical structures (LAM and TT).

33. The larger HRV statistical dispersion,

34. by better renal functioning (eGFR).

Reviewer #3: Authors conducted research concerning the autonomic response (changes of HRV) in acitve standing in patients post-kidney transplant. Differences in HRV reagibilty were historically compared with ESKD patients (on dialyis) without transplant. The small number (n=16), no direct statistical comparison to ESKD and the very young test population (age 32yr) stimulate doubts about general associations and the comparability to ESKD overall. However the methodology of HRV measurements is sound, the Ms. well written and undstandable. The reviewer recommends that these consideration, in particular the very young population should be mentioned and more critically discussed in the paper.

Reviewer #4: The article focuses on the HRV (heart rate variability) of patients after kidney transplantation. Impaired HRV in patients with ESRD is known to be improved by HD. The improvement of the sympathetic stimulation mediated HRV response in active standing after kidney transplantation is still unknown and the subject of the present study. The study question is very interesting and concisely explained in the introduction.

The question was investigated using frequency-based metrics, RIQA (recurrence quantitative analysis) and short term ECG. The methodology is explained conclusively and is well suited to the investigation of the question.

It could be shown significantly that HRV is lower in active standing in kidney transplant patients than in supine. Significant differences in various HRV indices were shown in active standing. However, the article points out that other studies in patients with ESRD, not hemodialysis patients, showed no significant difference in HRV in active standing. The results are explained conclusively.

A disadvantage is the small number of subjects (16). There is also a lack of data (HRV) on the same patients before transplantation, as already noted by the authors

6. PLOS authors have the option to publish the peer review history of their article (what does this mean?). If published, this will include your full peer review and any attached files.

Reviewer #1: No

Reviewer #2: **Yes: **ken berend

Reviewer #3: No

Reviewer #4: No

---

## [Author Response · Author response to Decision Letter 0]

28 Apr 2023

Manuscript PONE-D-23-02358 “Characterization of nonlinear heart rate variability of end-stage renal disease patients after kidney transplantation”

Response to reviewers

Reviewer #1

Comment: The study exploits recurrence quantification analysis (RQA) to characterize heart rate variability (HRV) of end-stage renal disease (ESRD) patients after kidney transplantation (KT).

The study is interesting and originally discuss HRV during orthostatic stimulus in ESRD patients after KT. However, some issues need a more profound discussion.

1) The major finding of the present study is the greater regularity of HRV during standing in ESRD patients after KT and this result supports the general improvement after KT. A greater regularity of HRV is compatible with an increased sympathetic modulation during orthostatic challenge as proven in A. Porta et al, J Appl Physiol, 103, 1143-1149, 2007. The link between loss of HRV complexity and postural challenge should be more deeply discussed and corroborated with adequate references.

Response: We thank the reviewer for the comments, which greatly help to improve the clarity and quality of our work. In the revised version of the manuscript, we discuss in greater detail the effects of active standing (page 10).

Comment: 2) It remains unclear why the authors used RQA instead of simpler nonlinear, and more widely utilized, descriptors of HRV irregularity such as markers based on conditional entropy.

Response: In an effort to broaden the scope of tools for HRV analysis, we explore the changes of HRV using recurrence plots, as it has been also proposed by other authors (10.1016/j.bspc.2015.10.007, 10.3389/fphys.2020.00040, 10.1371/journal.pone.0249504). However, in the new version of the manuscript we discuss the possibility of using other nonlinear methods, which may be or not simpler (page 12).

Comment: 3) The authors evoked the presence of nonlinear dynamics in HRV (e.g., in the title) but no tests were utilized to check the actual presence of nonlinear dynamics. Tests should be carried out to corroborate the choice of a nonlinear marker (see e.g., A. Porta et al, Front Physiol, 6, 71, 2015 for a procedure to test the presence of nonlinear dynamics). In alternative, linear markers of complexity are present (see A. Porta et al, IEEE Trans Biomed Eng, 64, 1287-1296, 2017).

Response: In the new version of the manuscript, we changed the title and abstract to fit the purpose of this study, which is to describe HRV by traditional and RQA measures in kidney transplantation, rather than the direct demonstration of a particular dynamical behavior through surrogate data testing. 

We acknowledge the usefulness of these tests, and we propose to test nonlinear behavior per se in a larger sample and make direct statistical test against HRV before kidney transplantation (Discussion section page 12).

Comment: 4) More details should be given to better understand how the parameters of RQA were chosen. For example, it is unclear how the embedding dimension and the tolerance to assess pattern similarity were selected. Please clarify in the Methods.

Response: In the revised version, we explain with more detail the selection of parameters (Methods section page 6), which is based on methodological research in recurrence plots (Marwan et al. 2007, reference 11).

First, we obtain the averaged histogram based mutual information of a given heart rate variability (HRV) time series, from which we obtain the first local minimum of the function. The delay at this local minimum corresponds to the time delay of the signal (tau). After a value for tau is chosen, we use it to compute the false nearest neighbors’ function to determine the dimension (m) for the recurrence plot construction. For the latest, we also take the first local minimum of the function (we provided an overview of this selection in a previous work – (reference 12).

Comment: 5) The white squares in the recurrence plots in Fig.1 denote the presence of nonstationarities. Their presence is visible as changes of the mean in the same figure. Since stationarity is a prerequisite of RQA, it should be checked (see V. Magagnin et al, Physiol Meas, 32, 1775-1786, 2011 for a stationarity test). In alternative, the authors should discuss the impact of nonstationarities over the final conclusion of the study.

Response: Indeed, it is qualitatively observed the nonstationary behavior of HRV. In a previous work (reference 15), we showed that RQA obtains pertinent information contained in HRV of healthy and ESRD patients, despite its nonstationary behavior (tested by the proposed algorithm). We added in the discussion the potential impact of nonstationarity in our results (page 12).

Comment: 6) English language should be profoundly revised (e.g., the second phrase from the bottom of the Abstract). Construction of the phrases should be checked (e.g., the first phrase of the Introduction) as well as the usage of punctuation.

Response: We reviewed thoroughly the language usage.

Reviewer #2

Comment: This paper concerns “Characterization of nonlinear heart rate variability of end-stage renal disease patients after kidney transplantation”

It is a struggle for me to read the paper because of many language issues and a huge number of abbreviations. I am not familiar with all these cardiology terms and abbreviations, so that does not help. Although the subject is interesting, after trying to read it several times, at present I cannot comment properly because of the numerous textual issues.

Table 2 abbreviations after normal text would make it more readable.

A list of abbreviations is also required

A clear legend of the figures is necessary

The following sentences need improvement.

1. However, it is unknown whether this response is recovered after definitive treatment with kidney transplantation.

2. We studied HRV dynamics by obtaining short-term ECG recordings from kidney transplant recipients that underwent active standing stimulation

3. Larger estimated glomerular filtration rate (eGFR) in significantly correlated with

broader gross HRV in supine position and during active standing. The loss in HRV during active standing may indicate a preserved sympathetic response to external stimuli

4. is known that chronic sympathetic hyperactivity produces decreased HRV modulation with predominance of lowfrequency oscillations, and a blunted HRV response to active standing

5. This tool allows to graphically represent the recurrence of a system to a particular dynamical state

6. This maneuver allows studying of the effects of volume redistribution and sympathetic

stimulation of the cardiovascular system

7. However a kidney transplant is the treatment of choice for ESRD

8. We included 16 patients with ESRD after kidney transplantation that were recruited during followup consultation in a second-level hospital in Mexico with a median age of 32 �27-35� years old

9. and triple immunosuppression treatment with a calcineurin inhibitor plus prednisone and mycophenolate.

10. Subjects were asked to avoid caffeinated beverages and other stimulants

11. We name the final HRV time series as NN

12. We calculate the mean of NN time series (meanNN

13. Embedding dimension was calculated at the first local minimum reaching zero in the fixed amount of neighbors function

14. Categorical variables are reported as absolute frequency (relative frequency) and continuous

variables as median (interquartile range).

15. Time domain indices, meanNN, SDNN, SDSD, and pNN20 are larger during supine position in comparison to active standing

16. Few correlations are observed in supine position (SDSD, pNN20, DET and ENT), but during active standing a significant correlation is found between eGFR and most of HRVmeasures (SDNN, SDSD, Pnn20, LF, HF, LF/HF, DET, LAM, TT and T2).

17. Also, a larger standard deviation of 5-minutes intervals in 24 hours (SDANN) of kidney transplant recipients without diabetes than those recipients without diabetes

18. Here, patients after kidney transplantation

19. However, we found here that in transplant recipients, these measures increase their value during active standing.

20. change in healthy subjects in response to active standing. given that the rapid changes in heart rate

21. is due to dysautonomia. This

22. other experimental scenarios [14, 18], which are not found in patients with ESRD but after HD treatment.

23. approach such be conducted

24. our study do not have

25. it is likely that the changes in RQA measures in active standing (e.g. DET, ENT, LAM and TT) may be a result of fluid overload

26. resistance [5, 25], these

27. As we show in this

28. is related to better eGFR

29. to a improve in HRV

30. to explore this relation and determine

31. in comparison with supine position

32. a consistent predominance of vertical structures (LAM and TT).

33. The larger HRV statistical dispersion,

34. by better renal functioning (eGFR).

Response: We thank the reviewer for the observations. In the revised version we added the meaning of abbreviations, clear legends to the figures, and a list of abbreviations, as requested. We also made corrections to the phrases that the reviewer highlighted, we show them below. Sentences with letter “a” correspond to the previous version; those with letter “b”, to the revised version.

1a. However, it is unknown whether this response is recovered after definitive treatment with kidney transplantation.

1b. However, it is unknown whether the response to active standing is recovered after definitive treatment with kidney transplantation (page 2).

2a. We studied HRV dynamics by obtaining short-term ECG recordings from kidney transplant recipients that underwent active standing stimulation

2b. We studied HRV dynamics by obtaining 5-minutes ECG recordings from kidney transplant recipients that underwent an active standing test (page 2).

3a. Larger estimated glomerular filtration rate (eGFR) in significantly correlated with

broader gross HRV in supine position and during active standing. The loss in HRV during active standing may indicate a preserved sympathetic response to external stimul

3b. A larger estimated glomerular filtration rate (eGFR) was significantly correlated with broader HRV in supine position and during active standing. The narrower HRV during active standing may indicate a recovered sympathetic response to external stimuli (page 2).

4a. is known that chronic sympathetic hyperactivity produces decreased HRV modulation with predominance of lowfrequency oscillations, and a blunted HRV response to active standing

4b. chronic sympathetic hyperactivity produces decreased HRV modulation with a predominance of low-frequency oscillations and a blunted HRV response to active standing (page 3).

5a. This tool allows to graphically represent the recurrence of a system to a particular dynamical state

5b. This tool graphically represents the recurrences of a system to a particular dynamical state (page 3).

6a. This maneuver allows studying of the effects of volume redistribution and sympathetic

stimulation of the cardiovascular system 

6b. This maneuver allows studying the effects of blood volume redistribution and sympathetic stimulation of the cardiovascular system (page 3)

7a. However a kidney transplant is the treatment of choice for ESRD

7b. However, a kidney transplant is the treatment of choice for ESRD (page 3).

8a. We included 16 patients with ESRD after kidney transplantation that were recruited during followup consultation in a second-level hospital in Mexico with a median age of 32 �27-35� years old

8b. We included 16 patients with ESRD after kidney transplantation recruited during follow-up consultation in a second-level hospital in Mexico with a median age of 32 [27-35] years old (page 4).

9a. and triple immunosuppression treatment with a calcineurin inhibitor plus prednisone and mycophenolate.

9b. triple immunosuppression regimen with a calcineurin inhibitor, prednisone and mycophenolate (page 4).

10a. Subjects were asked to avoid caffeinated beverages and other stimulants

10b. During recruitment, subjects were asked to avoid caffeinated beverages and other stimulants 24 hours before the evaluation (page 4).

11a. We name the final HRV time series as NN

11b. We named the final HRV time series as NN intervals (page 6).

12a. We calculate the mean of NN time series (meanNN

12b. We calculated the mean of NN time series (meanNN [s]), page 6.

13a. Embedding dimension was calculated at the first local minimum reaching zero in the fixed amount of neighbors function

13b. Embedding delay was chosen at the average mutual information function's first local minimum for each time series. The embedding dimension was calculated at the false nearest neighbors function (page 6).

14a. Categorical variables are reported as absolute frequency (relative frequency) and continuous variables as median (interquartile range).

14b. We report categorical variables as absolute frequency (relative frequency), and continuous variables as median (interquartile range), page 7.

15a. Time domain indices, meanNN, SDNN, SDSD, and pNN20 are larger during supine position in comparison to active standing

15b. The values of time domain indices, meanNN, SDNN, SDSD, and pNN20 are larger during supine position than active standing (Figure 2), page 7.

16a. Few correlations are observed in supine position (SDSD, pNN20, DET and ENT), but during active standing a significant correlation is found between eGFR and most of HRVmeasures (SDNN, SDSD, Pnn20, LF, HF, LF/HF, DET, LAM, TT and T2).

16b. Few correlations are observed between eGFR and HRV indices (SDSD, pNN20, DET and ENT) in supine position. During active standing, significant correlations are found between eGFR and most of HRV measures (SDNN, SDSD, Pnn20, LF, HF, LF/HF, DET, LAM, TT and T2), page 9.

17a. Also, a larger standard deviation of 5-minutes intervals in 24 hours (SDANN) of kidney transplant recipients without diabetes than those recipients without diabetes

17b. Also, a larger standard deviation of 5-minutes intervals in 24 hours has been observed in kidney transplant recipients without diabetes than those recipients with diabetes (page 10).

18a. Here, patients after kidney transplantation

18b. In this study, patients after kidney transplantation (page 10).

19a. However, we found here that in transplant recipients, these measures increase their value during active standing.

19b. However, we found here that these measures increase their value during active standing in transplant recipients (page 10).

20a. change in healthy subjects in response to active standing. given that the rapid changes in heart rate

20b. change in healthy subjects in response to active standing, given that the rapid changes in heart rate (page 11).

21a. is due to dysautonomia. This

21b. It has been suggested that the lack of a significant response in ESRD is due to autonomic nervous system impairment (dysautonomia). This (page 11).

22a. other experimental scenarios [14, 18], which are not found in patients with ESRD but after HD treatment.

22b. other experimental scenarios [15, 18], in which no correlation is found in patients with ESRD before HD, but such correlation is recovered after HD treatment.

23a. approach such be conducted

23b. approach should be conducted (page 11). 

24a. our study do not have

24b. our study does not have (page 11).

25a. it is likely that the changes in RQA measures in active standing (e.g. DET, ENT, LAM and TT) may be a result of fluid overload

25b. it is likely that the changes in RQA measures observed in active standing (e.g. DET, ENT, LAM and TT) may be the result of fluid overload (age 12)

26a. resistance [5, 25], these

26b. resistance [5, 26]. These (page 12)

27a. As we show in this

27b. In this (page 12).

28a. is related to better eGFR

28b. is related to higher eGFR (page 12)

29a. to a improve in HRV

29b. a improvement in HRV (page 12)

30a. to explore this relation and determine

30b. to explore this relationship and to determine (page 12)

31a. in comparison with supine position

31b. compared with the supine position (page 13).

32a. a consistent predominance of vertical struct

---

## [Decision Letter · Decision Letter 1]

25 May 2023

PONE-D-23-02358R1Characterization of heart rate variability in end-stage renal disease patients after kidney transplantation with recurrence quantification analysisPLOS ONE

Dear Dr. González-Gómez,

Thank you for submitting your manuscript to PLOS ONE. After careful consideration, we feel that it has merit but does not fully meet PLOS ONE’s publication criteria as it currently stands. Therefore, we invite you to submit a revised version of the manuscript that addresses the points raised during the review process.

We look forward to receiving your revised manuscript.

Kind regards,

Keiko Hosohata, Ph.D.

Academic Editor

PLOS ONE

Reviewers' comments:

Reviewer's Responses to Questions

**Comments to the Author**

1. If the authors have adequately addressed your comments raised in a previous round of review and you feel that this manuscript is now acceptable for publication, you may indicate that here to bypass the “Comments to the Author” section, enter your conflict of interest statement in the “Confidential to Editor” section, and submit your "Accept" recommendation.

Reviewer #5: All comments have been addressed

Reviewer #6: All comments have been addressed

2. Is the manuscript technically sound, and do the data support the conclusions?

Reviewer #5: Partly

Reviewer #6: Partly

3. Has the statistical analysis been performed appropriately and rigorously? 

Reviewer #5: Yes

Reviewer #6: I Don't Know

4. Have the authors made all data underlying the findings in their manuscript fully available?

Reviewer #5: Yes

Reviewer #6: Yes

5. Is the manuscript presented in an intelligible fashion and written in standard English?

Reviewer #5: No

Reviewer #6: Yes

6. Review Comments to the Author

Reviewer #5: The authors set out to assess the response of heart rate variability (HRV) linear indices and recurrence plot quantitative analysis (RQA) measures in an active standing test on kidney transplant recipients. They concluded that HRV of kidney recipients shows a diminished variability during active standing, in comparison with supine position. That the dynamic behavior of HRV also changed in response to active standing with a consistent predominance of vertical structures.They went on to suggest that these changes suggest a marked sympathetic response to active standing, and a preserved capacity to respond to such orthostatic challenges with appropriate adjustments in the dynamic behavior of the cardiovascular control system.

This is an interesting topic that definitely requires more study and I am appreciative of these authors efforts. The paper has a number of deficiencies:

1. It is hard to read. The writing in English could be better, shorter and more succinct. For instance, the first sentence either needs to be two sentences or have a semi-colon. "Heart rate variability (HRV) is the instantaneous change of heart rate, these variations of heart rate are tightly related to respiratory and autonomic modulation of the cardiovascular system." Consider splitting this into two sentences or use a semi-colon instead of the coma. The sentence " However a kidney transplant is the treatment of choice for ESRD over lifetime dialysis in most clinical scenarios" is also unnecessary since it is universally expected that kidney transplantation is the best treatment modality for patients with ESKD.

2. The authors suggest that kidney transplantation may be the cause of the "diminished variability during active standing, in comparison with supine position." This however is an observational study and we cannot draw causality from the results of this study.

3. The study population is small and young. The study population has only 16 patients and the average age was 32 years. The average age for patients with ESRD is much higher (45-60s when initiating dialysis in Mexico) and it would have been more interesting and applicable if the authors studied patients with reflect the general ESRD population. To be fair, other studies on HRV also had younger populations ( e.g. Martín Calderón-Juárez et al's paper had average age of 27+/- 8 years). In general, the paper's generalizability is limited.

Reviewer #6: Thank you for your manuscript.

As a nephrology reviewer I am unable to understand the multiple abbreviations. Not only that what these abbreviations stand for and their significance.

The authors need to clarify the following statements.

1.What do you mean ESRD with Kidney transplantation? Is this a case of allograft failure?

2.What is the implications of replacing heart beats in the final analysis. Is 5% accepted standards?

3.What is the final message from the study and is there anything new?

4. It would be instructive to know what anti-hypertensives that the transplant recipients were on and their impact on HRV

Improvement in autonomic activity with improvement of eGFR is a known phenomenon.

7. PLOS authors have the option to publish the peer review history of their article (what does this mean?). If published, this will include your full peer review and any attached files.

Reviewer #5: No

Reviewer #6: **Yes: **Urmila Anandh

---

## [Author Response · Author response to Decision Letter 1]

26 Jun 2023

We appreciated all the observations and questions made to our manuscript, all of which we are sure will improve our paper’s quality. Next, we will provide answers to the questions and observations made by the reviewers.

Reviewer #5

Comment: The authors set out to assess the response of heart rate variability (HRV) linear indices and recurrence plot quantitative analysis (RQA) measures in an active standing test on kidney transplant recipients. They concluded that HRV of kidney recipients shows a diminished variability during active standing, in comparison with supine position. That the dynamic behavior of HRV also changed in response to active standing with a consistent predominance of vertical structures. They went on to suggest that these changes suggest a marked sympathetic response to active standing, and a preserved capacity to respond to such orthostatic challenges with appropriate adjustments in the dynamic behavior of the cardiovascular control system.

This is an interesting topic that definitely requires more study and I am appreciative of these authors efforts. The paper has a number of deficiencies:

1. It is hard to read. The writing in English could be better, shorter and more succinct. For instance, the first sentence either needs to be two sentences or have a semi-colon. "Heart rate variability (HRV) is the instantaneous change of heart rate, these variations of heart rate are tightly related to respiratory and autonomic modulation of the cardiovascular system." Consider splitting this into two sentences or use a semi-colon instead of the coma. The sentence " However a kidney transplant is the treatment of choice for ESRD over lifetime dialysis in most clinical scenarios" is also unnecessary since it is universally expected that kidney transplantation is the best treatment modality for patients with ESKD.

Response: Thank you because of the observation. We reviewed all the text and made changes to improve the writing in English, as you will see in different sections of the paper. We agree with you regarding kidney transplant as the universally accepted best treatment choice in patients with end-stage renal disease. We removed the sentence. 

Comment: 2. The authors suggest that kidney transplantation may be the cause of the "diminished variability during active standing, in comparison with supine position." This however is an observational study and we cannot draw causality from the results of this study.

Response: We completely agree with you. Our study cannot establish causality in the improvement of heart rate variability resulting from kidney transplantation. 

As we mention on page 12, paragraph 2: “…our study does not have HRV measurements of these patients before kidney transplantation; hence, it cannot be determined whether kidney transplantation improves RQA measures”. In the same paragraph, we compare the present results with previous work on patients treated with hemodialysis. At the end, we mention that: “A better eGFR may be linked to an improvement in HRV measures as observed in our work”. However, we recognize that further studies are required to assess if this relationship is related to the improvement of cardiovascular function due to kidney transplantation. 

To avoid confusion in our Conclusions section, we included a paragraph emphasizing that further studies are required to demonstrate causality.

Comment: 3. The study population is small and young. The study population has only 16 patients and the average age was 32 years. The average age for patients with ESRD is much higher (45-60s when initiating dialysis in Mexico) and it would have been more interesting and applicable if the authors studied patients with reflect the general ESRD population. To be fair, other studies on HRV also had younger populations (e.g. Martín Calderón-Juárez et al's paper had average age of 27+/- 8 years). In general, the paper's generalizability is limited.

Response: We agree that the patients included in our study are very young, and we understand that this situation may be representative of most patients undergoing kidney transplantation in your country, with a mean age of 29 years. However, we also acknowledge that national statistics can vary, and the global reality may present a different situation regarding the age of patients with end-stage chronic kidney disease. Our study expanded the discussion to address this point and provide a more comprehensive understanding of the situation. 

Reviewer #6

Comment: Thank you for your manuscript.

As a nephrology reviewer I am unable to understand the multiple abbreviations. Not only that what these abbreviations stand for and their significance.

Response: The most recent manuscript includes a list of abbreviations. 

Comment: The authors need to clarify the following statements.

1.What do you mean ESRD with Kidney transplantation? Is this a case of allograft failure?

Response: We agree with you; it can be confusing. The patients included in the study had evidently experienced end-stage renal disease and received dialysis therapy at some point. At the time of their inclusion in the study, they had a functional kidney transplant. The sentence has been modified to avoid confusion (page 4, line 1).

Comment: 2.What is the implications of replacing heart beats in the final analysis. Is 5% accepted standards?

Response: According to the guidelines for heart rate variability analysis (Reference 1), the analysis should involve only RR intervals measured during sinus rhythm only. Therefore, RR intervals that involve extrasystoles should not be included. The guidelines indicate that up to 5% of replaced heartbeats are acceptable. The reference to guidelines is now on page 6, paragraph 1.

Comment: 3.What is the final message from the study and is there anything new?

Response: As mentioned in the Conclusions section, our work demonstrated that kidney recipients show a preserved physiological response to an active orthostatic challenge. The response is consistent with increased sympathetic activity during orthostasis. Moreover, the recurrence plot analysis also evidenced interesting changes in the dynamic behavior of HRV.

Comment: 4. It would be instructive to know what anti-hypertensives that the transplant recipients were on and their impact on HRV. 

Response: Indeed. We included that information in the article (page 4, paragraph 1). Only three patients were on antihypertensive medications, of whom only one was taking a low-dose non-cardioselective beta-blocker.

Comment: Improvement in autonomic activity with improvement of eGFR is a known phenomenon.

Response: Correct. However, the study of autonomic activity is not straightforward. Here we used a non-invasive approach with heart rate variability during a controlled physiological maneuver (active standing). The response of the cardiac autonomic modulation to this maneuver is well known for healthy and end-stage renal patients treated by hemodialysis. Our work shows that kidney transplant recipients have a preserved autonomic response to such a maneuver. Also, nonlinear indices of heart rate variability have emerged as a helpful tool to analyze this, and that is what we explored in this study. Our results showed that the correlations between eGFR and heart rate variability indices were more evident for most indices when the physiological stress of active standing was applied.

---

## [Decision Letter · Decision Letter 2]

15 Sep 2023

PONE-D-23-02358R2Characterization of heart rate variability in end-stage renal disease patients after kidney transplantation with recurrence quantification analysisPLOS ONE

Dear Dr. González-Gómez,

Thank you for submitting your manuscript to PLOS ONE. After careful consideration, we feel that it has merit but does not fully meet PLOS ONE’s publication criteria as it currently stands. Therefore, we invite you to submit a revised version of the manuscript that addresses the points raised during the review process.

We look forward to receiving your revised manuscript.

Kind regards,

Keiko Hosohata, Ph.D.

Academic Editor

PLOS ONE

Journal Requirements:

1.Please review your reference list to ensure that it is complete and correct. If you have cited papers that have been retracted, please include the rationale for doing so in the manuscript text, or remove these references and replace them with relevant current references. Any changes to the reference list should be mentioned in the rebuttal letter that accompanies your revised manuscript. If you need to cite a retracted article, indicate the article’s retracted status in the References list and also include a citation and full reference for the retraction notice.

Reviewers' comments:

Reviewer's Responses to Questions

**Comments to the Author**

1. If the authors have adequately addressed your comments raised in a previous round of review and you feel that this manuscript is now acceptable for publication, you may indicate that here to bypass the “Comments to the Author” section, enter your conflict of interest statement in the “Confidential to Editor” section, and submit your "Accept" recommendation.

Reviewer #1: All comments have been addressed

Reviewer #6: All comments have been addressed

2. Is the manuscript technically sound, and do the data support the conclusions?

Reviewer #1: Yes

Reviewer #6: Partly

3. Has the statistical analysis been performed appropriately and rigorously? 

Reviewer #1: Yes

Reviewer #6: I Don't Know

4. Have the authors made all data underlying the findings in their manuscript fully available?

Reviewer #1: Yes

Reviewer #6: Yes

5. Is the manuscript presented in an intelligible fashion and written in standard English?

Reviewer #1: Yes

Reviewer #6: Yes

6. Review Comments to the Author

Reviewer #1: The manuscript has been improved. The authors replied satisfactorily to all my issues and took into account the suggestions given. I have no additional comments.

Reviewer #6: The authors have addressed the comments raised by the reviewers.

Only concern is that the manuscript has many technical issues which are beyond the scope of transplant physicians. This manuscript is written for the cardiology sub-speciality audience.

For a nephrologist the fact that that the autonomic dysfunction improves post transplantation is well known.

It would have been more instructive if an attempt was made if the changes are merely attributed to improvement of GFR or other factors like improving anemia, PTH levels, electrolytes etc are also responsible.

7. PLOS authors have the option to publish the peer review history of their article (what does this mean?). If published, this will include your full peer review and any attached files.

Reviewer #1: No

Reviewer #6: No

---

## [Author Response · Author response to Decision Letter 2]

2 Oct 2023

Characterization of heart rate variability in end-stage renal disease patients after kidney transplantation with recurrence quantification analysis

Response to reviewers

Reviewer #6: The authors have addressed the comments raised by the reviewers.

Only concern is that the manuscript has many technical issues which are beyond the scope of transplant physicians. This manuscript is written for the cardiology sub-speciality audience.

For a nephrologist the fact that that the autonomic dysfunction improves post transplantation is well known.

It would have been more instructive if an attempt was made if the changes are merely attributed to improvement of GFR or other factors like improving anemia, PTH levels, electrolytes etc are also responsible.

Response: We thank the reviewer for their comments, which contributed to the quality improvement of the present manuscript. We acknowledge that the complex physiology behind eGFR and HRV might be influenced by a large and intricated number factors. The exploratory work that we present will contribute as a precedent to propose further research to help elucidate the mechanism and clinical implications of the correlation between HRV and eGFR. We address this issue in the revised version of the manuscript (Discussion section, page 12, paragraph 1, lines 11- 18).

---

## [Decision Letter · Decision Letter 3]

2 Jan 2024

PONE-D-23-02358R3Characterization of heart rate variability in end-stage renal disease patients after kidney transplantation with recurrence quantification analysisPLOS ONE

Dear Dr. González-Gómez,

Thank you for submitting your manuscript to PLOS ONE. After careful consideration, we feel that it has merit but does not fully meet PLOS ONE’s publication criteria as it currently stands. Therefore, we invite you to submit a revised version of the manuscript that addresses the points raised during the review process.

We look forward to receiving your revised manuscript.

Kind regards,

Choon-Hian Goh, Ph.D.

Academic Editor

PLOS ONE

Journal Requirements:

Reviewers' comments:

Reviewer's Responses to Questions

**Comments to the Author**

1. If the authors have adequately addressed your comments raised in a previous round of review and you feel that this manuscript is now acceptable for publication, you may indicate that here to bypass the “Comments to the Author” section, enter your conflict of interest statement in the “Confidential to Editor” section, and submit your "Accept" recommendation.

Reviewer #1: All comments have been addressed

Reviewer #6: All comments have been addressed

2. Is the manuscript technically sound, and do the data support the conclusions?

Reviewer #1: Yes

Reviewer #6: Partly

3. Has the statistical analysis been performed appropriately and rigorously? 

Reviewer #1: Yes

Reviewer #6: Yes

4. Have the authors made all data underlying the findings in their manuscript fully available?

Reviewer #1: Yes

Reviewer #6: Yes

5. Is the manuscript presented in an intelligible fashion and written in standard English?

Reviewer #1: Yes

Reviewer #6: Yes

6. Review Comments to the Author

Reviewer #1: The manuscript has been improved. The authors replied satisfactorily to all my issues and took into account the suggestions given. I have no additional comments.

Reviewer #6: The revised manuscript attempts to address most of the issues .

The central concern about the relevance to the nephrology community still remains.

7. PLOS authors have the option to publish the peer review history of their article (what does this mean?). If published, this will include your full peer review and any attached files.

Reviewer #1: No

Reviewer #6: No

---

## [Author Response · Author response to Decision Letter 3]

24 Jan 2024

Comment from the Editor (Journal Requirements)

Response: Thank you for this indication. We reviewed the reference list to verify that is complete and correct. We found that only one reference (number 25) had been retracted recently (November 2023):

25. Shi W, Zhang J, Chen D, Chen X, Duan W, Zhang H. Heart Rate Variability and Chronic Kidney Disease in Patients with Type 2 Diabetes. Appl Bionics Biomech. 2022;2022:2475750. Epub 20220517. doi:10.1155/2022/2475750. PubMed PMID: 35619730; PubMed Central PMCID: PMCPMC9129959.

We eliminated both the reference and the following phrase from page 12, paragraph 1: “In another study in which HRV was evaluated within 24 hours recordings, the correlation between SDNN and eGFR in patients with chronic kidney disease and type 2 diabetes was not significant [25].”

We also updated the order for the following references:

23. García-García G, García-Bejarano H, Breien-Coronado H, Perez-Cortez G, Pazarin-Villaseñor L, de la Torre-Campos L, et al. Chapter 9 - End-Stage Renal Disease in Mexico1. In: García-García G, Agodoa LY, Norris KC, editors. Chronic Kidney Disease in Disadvantaged Populations: Academic Press; 2017. p. 77-83.

24. Lerma C, Minzoni A, Infante O, Jose MV. A mathematical analysis for the cardiovascular control adaptations in chronic renal failure. Artif Organs. 2004;28(4):398-409. Epub 2004/04/16. doi: 10.1111/j.1525-1594.2004.47162.x. PubMed PMID: 15084202. 

25. Tahrani AA, Dubb K, Raymond NT, Begum S, Altaf QA, Sadiqi H, et al. Cardiac autonomic neuropathy predicts renal function decline in patients with type 2 diabetes: a cohort study. Diabetologia. 2014;57(6):1249-56. doi: 10.1007/s00125-014-3211-2.

Comments from the Reviewers

6. Review Comments to the Author

Reviewer #1: The manuscript has been improved. The authors replied satisfactorily to all my issues and took into account the suggestions given. I have no additional comments.

Reviewer #6: The revised manuscript attempts to address most of the issues.

The central concern about the relevance to the nephrology community still remains.

Response: We addressed previously this concern, specifically about the correlation between eGFR and HRV indices. As we mentioned in the previous round, such correlation is already known among nephrologists, as is cited in the manuscript (page 12, lines 1 to 3): “Correlation between greater HRV and better eGFR has been observed in patients with chronic kidney disease in a broad population with different autonomic maneuvers [25] .”

Moreover, as we mentioned in the previous round, the factors that may explain such correlation are diverse and not fully elucidated. In our manuscript, we discuss some of these clinically relevant factors (page 12, lines 11 to 18). Furthermore, as we mention in the same paragraph: “In this exploratory study, such factors were not evaluated, and further research is needed to assess their specific role with HRV modulation and eGFR.” 

We have added the following sentence in the same paragraph to point out the relevance of our findings:

“Since our approach of non-linear HRV indices obtained by recurrence plot analysis during a physiological stimulus (active standing) has been useful to address the dynamic adaptability in ESRD patients treated by HD [5,7], the current findings of the correlation between eGFR and RQA measures could motivate future investigation about the adaptability of the cardiovascular control mechanisms in transplanted patients.”

7. PLOS authors have the option to publish the peer-review history of their article (what does this mean?). If published, this will include your full peer review and any attached files.

Response: Yes

---

## [Editor Report · Decision Letter 4]

6 Feb 2024

Characterization of heart rate variability in end-stage renal disease patients after kidney transplantation with recurrence quantification analysis

PONE-D-23-02358R4

Dear Dr. González-Gómez,

We’re pleased to inform you that your manuscript has been judged scientifically suitable for publication and will be formally accepted for publication once it meets all outstanding technical requirements.

Kind regards,

Choon-Hian Goh, Ph.D.

Academic Editor

PLOS ONE